# Brixia Score in Outcomes of Alpha versus Delta Variant of Infection in Pregnant Critical COVID-19 Patients

Roxana Covali [1,*] , Demetra Socolov [2], Ioana Pavaleanu [3], Mona Akad [4], Lucian Vasile Boiculese [5] and Razvan Socolov [3]

1   Department of Radiology, Elena Doamna Obstetrics and Gynecology University Hospital, Grigore T. Popa University of Medicine and Pharmacy Iasi, 700115 Iasi, Romania
2   Department of Obstetrics and Gynecology, Cuza Voda Obstetrics and Gynecology University Hospital, Grigore T. Popa University of Medicine and Pharmacy Iasi, 700115 Iasi, Romania; socolov@hotmail.com
3   Department of Obstetrics and Gynecology, Elena Doamna Obstetrics and Gynecology University Hospital, Grigore T. Popa University of Medicine and Pharmacy Iasi, 700115 Iasi, Romania; ioana-m-pavaleanu@umfiasi.ro (I.P.); socolovr@yahoo.com (R.S.)
4   Department of Obstetrics and Gynecology, Grigore T. Popa University of Medicine and Pharmacy, 700115 Iasi, Romania; akad.mona@yahoo.com
5   Department of Statistics, Grigore T. Popa University of Medicine and Pharmacy Iasi, 700115 Iasi, Romania; lboiculese@gmail.com
*   Correspondence: rcovali@yahoo.com; Tel.: +40-232-210-390 (ext. 275)

**Abstract:** Background: Critical COVID-19 patients account for 1.7 to 13% of all pregnant COVID-19 patients. Methods: Patients admitted to the COVID-19 intensive care unit of Elena Doamna Obstetrics and Gynecology University Hospital in Iasi between 1 January and 1 December 2021, with critical forms of the disease, were included and retrospectively studied. The patients' age range was 25–44 years in the Alpha group (n = 12) and 27–52 years in the Delta group (n = 9). Results: Most critically ill pregnant COVID-19 patients in the Alpha group delivered when admitted to the intensive care unit, while less than half of those in the Delta group delivered when admitted; the rest were released home and continued their pregnancy normally. There was a significant difference regarding the number of patients released to home care and the number of days after admission when delivery occurred ($p$ = 0.02 and 0.022, respectively). Conclusions: There was no significant difference in maternal and fetal outcomes between the two groups, except for the number of patients released to home care and the number of days after admission when delivery occurred. There was no correlation between any Brixia scores (H, L, A, E) and any maternal or fetal outcomes in both groups.

**Keywords:** COVID-19; pregnancy; Brixia score; maternal outcomes; fetal outcomes; cesarean delivery; Alpha variant; Delta variant

## 1. Introduction

Severe pulmonary disease caused by the novel coronavirus (severe acute respiratory syndrome coronavirus 2 (SARS-CoV-2)) has devastated many countries around the world and has overwhelmed the medical system [1]. The priorities of many institutions have changed to manage critically ill coronavirus disease 2019 (COVID-19) patients [2].

COVID-19 represents a challenge in balancing maternal and fetal interests in terms of reducing morbidity and mortality; therefore, there is often uncertainty in clinical judgment about the timing and mode of delivery [3]. Pregnant patients admitted to the intensive care unit (ICU) constitute a relatively unique population in that both the maternal and fetal status must be taken into account simultaneously [4]. Critical COVID-19 patients account for 1.7% [5,6] to 13% [7–10] of all pregnant COVID-19 patients.

In one study, Eman [11] reported an ICU mortality rate of 42.1%, representing all eight COVID-19 pregnant patients who were intubated and mechanically ventilated. Lokken [3] and Moieindarbary [12] reported only one patient admitted to the ICU, who subsequently

died of respiratory failure after a prolonged ICU stay despite multiple COVID-19 therapeutics, which is a 100% mortality rate. Lucovnik [6] reported a 33.33% mortality rate in critically ill pregnant patients. The World Association of Perinatal Medicine [13] reported 3 maternal deaths among 36 patients admitted to the ICU, which is an 8.33% mortality rate.

Given the anticipated favorable neonatal outcomes, Rose [4] suggested that for pregnant patients who require advanced oxygen delivery modalities, elective delivery should be considered at the gestational age of 32 to 33 weeks. DeBolt [8] reported that among pregnant women with critical coronavirus disease who underwent delivery, 72.7% had a cesarean section, and the mean gestational age at delivery was $35 \pm 3.5$ weeks. Polcer [14] reported a gestational age of 21–40 weeks at ICU admission, no deliveries, and no deaths. Panagiotakopoulos [15] reported an 88% rate of preterm delivery exclusively in the critically ill population. Pierce Williams [16] reported that 94% of deliveries were by cesarean section.

According to Maroldi [17], the Brixia score correlates strongly with disease severity and outcome; it can support clinical decision making, particularly for patients with moderate to severe signs and symptoms. Sargent [18], who described a scoring system stratifying X-rays by severity and directly linking this to prognosis, was unable to demonstrate this association in patients commencing CPAP treatment.

Since there is no agreement among studies regarding pregnant critical COVID-19 patients and their evolution, the need for another study analyzing the outcomes of these patients was apparent.

## 2. Materials and Methods

Patients admitted to the COVID-19 intensive care unit (ICU) of Elena Doamna Obstetrics and Gynecology University Hospital in Iasi between 1 January and 1 December 2021, who had a critical form of the disease and underwent conventional chest X-rays, were included and retrospectively studied. Most of them were pregnant patients (n = 18), and few were non-pregnant female patients (n = 3). Patients who had mild or moderate forms of the disease and patients who were in the COVID-19 ICU because they had just been operated upon and were also positive for COVID-19 (RT-PCR test) were excluded from the study. Patients who were in the ICU because they had just delivered by cesarean section, but were negative for COVID-19, were also excluded from this study. Patients were classified as having a critical form of COVID-19 if they had respiratory failure requiring endotracheal intubation, shock, or other organ failure that required intensive care [19,20].

There were no critical COVID-19 patients who did not have at least one chest X-ray during the ICU hospitalization. Our hospital has no computed tomography machine, so chest X-ray was the only imaging tool available. We used a Siemens Polymobil 10 mobile X-ray device. Since most patients were in the ICU, barely breathing, the X-rays were taken in an intermediate position between lying and orthostatic, and a deep breath before taking the image could not be obtained voluntarily.

Patients were confirmed to have COVID-19 infection by an RT-PCR test performed in our hospital upon admission, or by another health facility before being sent to us. The RT-PCR tests collected from the patients were sent to the local infectious disease hospital, where they were interpreted, and the results came back 12 h later in the internal internet network and were available only to certain members of the medical staff, with coded access. Viral strains were determined by genomic sequences of viral samples, performed by the National Institute of Public Health nationwide and released as weekly bulletins [21].

We were a COVID-19 support hospital, and one part of our hospital was adapted and dedicated to COVID-19 patients, including a small ICU. Since our hospital is a university obstetrics and gynecology hospital, we could only admit female obstetrical and gynecological patients.

There were two groups: group 1 (spring group, admitted in March–May 2021), separated by a three-month interval from group 2 (autumn group, admitted in August–

November 2021). During the three-month interval, our hospital was free of critical cases of COVID-19.

In Romania, in spring 2021, the Alpha variant (20I, V1) massively predominated, and in autumn 2021, the Delta variant (21A, 21I, 21J) [22] massively predominated; therefore, in this paper, we considered that group 1 (spring group) was infected with the Alpha variant, and group 2 (autumn group) was infected with the Delta variant.

Since this was a retrospective study, written informed consent was waived. The study was approved by the Elena Doamna Obstetrics and Gynecology University Hospital Research Ethics Committee, and the study was performed in accordance with the ethical standards as laid down in the 1964 Declaration of Helsinki and its later amendments (Number 4, April 2, 2020). This was part of a larger study on COVID-19 patients that started in April 2020 and is still ongoing in our hospital.

Statistical analysis was performed with SPSS version 18 software (SPSS Inc., Chicago, IL, USA). For descriptive measures, we computed the mean, standard deviation, median, and quartiles 1 and 3 (because the variables follow a non-normal distribution). Therefore, to compare the data, the nonparametric Mann–Whitney U test was applied. The standard significance cut-off at $p = 0.05$ was used to determine the hypothesis conclusion.

## 3. Results

The patients' age range was 25–44 years in group 1 (Alpha, spring group) and 27–52 years in group 2 (Delta, autumn group). The mean age and percentage of pregnant patients were not significantly different between the two groups ($p = 0.156$ and 0.553, respectively) (Table 1). Both groups consisted of mostly pregnant patients and a few non-pregnant patients (Table 1).

**Table 1.** Patient characteristics: mean values (±standard deviation) on the upper line, and median values (quartile 1, quartile 3) on the line below.

| Patients | Group 1 (Alpha Variant) (n = 12) | Group 2 (Delta Variant) (n = 9) | *p*-Value |
|---|---|---|---|
| Age | 32.75 (±6.26) 31.50 (27.5, 37.00) | 36.56 (±7.19) 37.00 (31.50, 39.50) | 0.175 |
| Pregnant patients | 11 (91.66%) | 7 (77.77%) | 0.553 |

The Brixia score was evaluated on the chest X-rays as follows: each lung was divided into three regions, from the apex to the base, and changes in the lungs were noted for each of the six regions as follows: normal (0), interstitial changes (1), interstitial and alveolar changes with interstitial predominance (2), and interstitial and alveolar changes with alveolar predominance (3). The resulting total Brixia score could be from 0 to 18. [15]. We studied the highest score (H-score), corresponding to the only score for patients who had only one chest X-ray and the highest score for those with multiple X-rays, and the lowest score (L-score), the score assigned at admission (A-score), and the last score obtained before hospitalization ended (E-score) or the patient died [15]. All X-rays were evaluated and scored by two independent investigators (RC and RS) blinded to the clinical data. Any differences were reconciled. Interobserver agreement was 0.94.

The Brixia score in the two groups varied as follows (Table 2): for patients with the Alpha variant, there was slightly higher lung involvement on admission in spring than in autumn, and they were released home or sent to another hospital with slightly less lung involvement than patients with the Delta variant; still, there was no significant difference between these scores (Table 2).

**Table 2.** Brixia score in the two groups of patients: mean values (±standard deviation) on the upper line, and median values (quartile 1, quartile 3) on the line below.

| Patients' Brixia Score | Group 1 (n = 12) | Group 2 (n = 9) | *p*-Value |
|---|---|---|---|
| Brixia H-score | 11.50 (±5.50) 13.00 (8.00, 16.00) | 11.67 (±3.16) 12.00 (9.50, 14.00) | 0.72 |
| Brixia L-score | 7.14 (±3.23) 6.00 (5.00, 10.00) | 7.00 (±5.35) 7.50 (1.75, 11.75) | 0.77 |
| Brixia A-score | 11.57 (±4.39) 13.00 (7.00, 15.00) | 10 (±6.97) 11.50 (2.75, 15.75) | 1 |
| Brixia E-score | 9.00 (±4.24) 8.00 (5.00, 13.00) | 8.50 (±5.68) 11.00 (2.75, 11.75) | 0.84 |

H-score: highest score, corresponding to only score for patients with only one chest X-ray and highest score for those with multiple X-rays; L-score: lowest score; A-score: score assigned at admission; E-score: last score obtained before hospitalization ended or patient died.

Though there was no significant difference in the Brixia scores between the two groups, there was a significant difference in outcomes between the two groups: all patients were discharged to home care in group 2 (Delta), while only one-third were discharged to home care in group 1 (Alpha), *p* = 0.02. There was a significant difference (*p* = 0.022) between the two groups regarding the number of days from hospital admission until delivery. In addition, most patients infected with the Alpha variant delivered when critically ill, while only less than half of those infected with the Delta variant delivered when critically ill, and the rest continued their pregnancy normally. Therefore, the Brixia score could not be used to predict the outcomes.

### 3.1. Lung Involvement

Peculiarities in lung involvement in spring versus autumn are shown in Table 3, showing no significant differences between the two groups. There was bilateral lung involvement in most cases in both groups, and there was no definite delineation between central and peripheral lung involvement in most patients. Pleural effusions were extremely rare.

**Table 3.** Lung involvement in the two groups of patients.

| Patients' Lung Involvement | Group 1 (n = 12) | Group 2 (n = 9) | *p*-Value |
|---|---|---|---|
| Bilateral lung involved | 11 (91.67%) | 9 (100%) | 0.375 |
| Left lung involved | 1 (8.33%) | 0 (0%) | 0.375 |
| Central distribution of opacities | 1 (8.33%) | 0 (0%) | 0.375 |
| Peripheral distribution of opacities | 4 (33.33%) | 6 (66.67%) | 0.130 |
| Neither central nor peripheral | 7 (58.33%) | 3 (33.33%) | 0.256 |
| Right pleural effusion | 1 (8.33%) | 0 (0%) | 0.375 |

### 3.2. Patients' Evolution

The evolution of patients can be seen in Table 4. The percentage of patients requiring intubation and the days of hospitalization were not significantly different in the two groups. There was a significant difference between the Alpha and Delta variants: in autumn, most patients were sent to home care and respiratory rehabilitation, while in spring, only one-third of patients were sent home, while two-thirds were sent to other higher-ranking hospitals. This evolution reflects that either the Alpha variant was more aggressive than Delta in the absence of available vaccines or, most probably, there was huge progress in medical staff experience and the availability of medications for critical cases of COVID-19 between spring and autumn 2021.

**Table 4.** Clinical evolution in the two groups of patients. For days of hospitalization, mean values (and standard deviation) are on the upper line, and median values (quartiles 1 and 2) are on the line below.

| Patient Course | Group 1 (n = 12) | Group 2 (n = 9) | *p*-Value |
|---|---|---|---|
| Intubated | 3 (25%) | 2 (22.22%) | 0.882 |
| Days in hospital | 9.33 ($\pm$7.84) | 13.44 ($\pm$6.30) | 0.117 |
| | 6.50 (8, 4) | 14 (9, 19) | |
| Discharged to home care | 4 (33.33%) | 9 (100%) | 0.02 |
| Sent to infectious diseases ICU | 4 (33.33%) | 0 (0%) | 0.054 |
| Sent to pulmonary ICU | 3 (25%) | 0 (%) | 0.105 |
| Sent to nephrology ICU | 1(8.33%) | 0 (0%) | 0.375 |

### 3.3. Delivery

There was a difference between the two groups of critical patients: though most patients infected with the Alpha variant delivered when critically ill, less than half of the patients infected with the Delta variant delivered when critically ill, and the rest continued their pregnancy normally. Most patients delivered by cesarean section (Table 5).

**Table 5.** Infant delivery in the two groups of patients.

| Pregnant Patients | Group 1 (n = 11) | Group 2 (n = 7) | *p*-Value |
|---|---|---|---|
| Delivered in our hospital | 9 (81.82%) | 3 (42.86%) | 0.087 |
| Proportion by cesarean section | 7 (77.78%) | 3 (100%) | 0.371 |

### 3.4. Gestational Age at Delivery and Day of Delivery

Though the patients delivered at similar gestational ages, there was a significant difference between the two groups regarding the number of days from hospital admission to delivery (Table 6).

**Table 6.** Gestational age at delivery and days from admission to delivery for the two groups of patients: mean values (and standard deviation) on the upper line, and median values (quartiles 1 and 2) on the line below.

| Pregnant Patients Who Gave Birth | Group 1 (n = 9) | Group 2 (n = 3) | *p*-Value |
|---|---|---|---|
| Gestational age at delivery | 32.33 ($\pm$3.50) | 34.33 ($\pm$3.51) | 0.353 |
| | 32 (30, 35) | 34 (33, 36) | |
| Days from admission to delivery | 1.56 ($\pm$1.01) | 6.67 ($\pm$4.16) | 0.022 |
| | 1 (1, 3) | 8 (4, 9) | |

### 3.5. Neonate

The condition of neonates at birth, reflected by the Apgar score, was approximately similar in the spring and autumn groups, and birth weight was slightly higher in the autumn group, but without significant differences between the two groups (Table 7).

**Table 7.** Neonate outcome in the two groups of patients: mean values (and standard deviation) on the upper line, and median values (quartiles 1 and 2) on the line below.

| Neonates | Group 1 (n = 9) | Group 2 (n = 3) | *p*-Value |
|---|---|---|---|
| Alive | 8 (88.89%) | 3 (100%) | 0.546 |
| Male gender | 4 (44.44%) | 2 (66.67%) | 0.505 |
| Weight | 2.03 ($\pm$0.87) | 2.72 ($\pm$1.20) | 0.405 |
| | 1.90 (1.50–2.59) | 2.22 (1.85–3.50) | |
| Apgar score | 5.44 ($\pm$3.12) | 6 ($\pm$3.46) | 0.637 |
| | 6 (5, 7) | 8 (3.5, 8) | |

### 3.6. Maternal Comorbidities

Though maternal comorbidities were more numerous in the autumn than in the spring group (Table 8) and outcomes were slightly better in the autumn group, there was no significant difference between the two groups.

**Table 8.** Maternal comorbidities in the two groups of patients.

| Comorbidities | Group 1 (n = 12) | Group 2 (n = 9) | *p*-Value |
|---|---|---|---|
| Thyroid | 1 (8.33%) | 4 (44.44%) | 0.055 |
| Heart | 2 (16.67%) | 0 (0%) | 0.198 |
| Kidney | 2 (16.67%) | 4 (44.44%) | 0.163 |
| Liver | 3 (25%) | 3 (33.33%) | 0.676 |
| High blood pressure | 1 (8.33%) | 0 (0%) | 0.375 |
| Obesity | 0 (0%) | 11.11% | 0.237 |

### 3.7. Correlations

There was no correlation between any of the Brixia scores (H, L, A, E) and any of the maternal or fetal outcomes in both groups (correlation coefficient significance >0.05 in all cases).

## 4. Discussion

COVID-19 primarily affects the lungs, by alveolar and capillary epithelial destruction, interstitial fibrous proliferation, and pulmonary consolidation [23,24]. The virus generates desquamation of pneumocytes and hyaline membrane formation, pulmonary edema, and interstitial mononuclear inflammatory infiltrates dominated by lymphocytes, and all of these changes can generally be found in both lungs [25]. Thrombosis in the blood vessels, local hemorrhage, necrosis, and hemorrhagic infarction can occur in the lung tissue [26].

Chest CT is highly recommended as the preferred diagnostic imaging method for COVID-19 due to its high density and high spatial resolution. It can show multiple segmental ground-glass opacities, distributed predominantly in extrapulmonary/subpleural zones and along bronchovascular bundles, with a crazy-paving pattern and interlobular septal thickening and consolidation, while pleural effusion or mediastinal lymphadenopathy is rarely seen [27].

Radiologically, COVID-19 progresses rapidly, from no abnormal change or bronchitis in the early stage to multiple patchy opacities in the central/peripheral zones of the lungs, generally involving both lungs [28]. In critical cases, multiple diffusive consolidations in both lungs, appearing as "white lungs" with a small amount of pleural effusion, may occur [18]. Manifestations in the absorption stage include dissipation or decreased density of previous lesions, or evolution into fiber- or cord-like opacities [27,29]. Barisione [30] correlated histologic patterns and chest images: the early/exudative phase was associated with ground-glass opacity, mid/proliferative lesions with crazy paving, and the late/fibrous phase with the consolidation pattern, more frequently seen in the lower/middle lobes.

In primary hospitals without CT, or for severe and critically severe patients, conventional radiology is critical in assessing the lungs. Initial chest X-ray is also a useful tool for triaging those patients who might have poor outcomes with suspected COVID-19 infection and would benefit most from hospitalization [31]. Balbi [32] showed that in conventional radiology, ground-glass opacity with consolidation was the most common finding.

Various methods of assessing severely affected lungs with conventional chest X-rays have been described. Batra [33] introduced a scoring system that divides each lung into three zones (upper, middle, and lower) and evaluates opacity localization, and then divides each lung field into four quadrants and evaluates severity, opacity characteristics, and quality. It is a very detailed and precise scoring system, but time-consuming.

A severity score was used by Kim [31], in which the initial chest X-ray was graded on a scale of 0–3: grade 0, no alveolar opacities; grade 1, alveolar opacities in <1/3 of the

lung; grade 2, alveolar opacities in 1/3 to 2/3 of the lung; and grade 3, alveolar opacities in >2/3 of the lung. Although it is quick, it is not very precise.

An intermediate method [18,34] was used to divide the lung image into four quadrants, by the spine and the level of the carina, and each quadrant was assessed for extent of "hazy" or "dense" opacification, where "dense" was defined as sufficient to obscure the anterior rib margin, and each quadrant was scored. This method is quick, but still quite subjective.

We chose the Brixia scoring system, which is quick and precise.

The Brixia score, age, and cardiovascular disease were found to predict death [32,35]. Maroldi [17] found that H-score was significantly higher in deceased patients compared to discharged patients, and in patients with more than two conventional chest X-rays, A-, L-, and E-scores correlated significantly with outcome. This was not true in our case, as we found no significant difference between H-scores in the two groups; still, there was a significant difference between outcomes, as all patients in group 2 were discharged to home care due to progress in COVID-19 medical care.

There were protocol changes [36] and huge progress in the clinical care of severe COVID-19 patients during the year 2021. In patients hospitalized with COVID-19, the use of dexamethasone resulted in lower 28-day mortality among those who were receiving either invasive mechanical ventilation or oxygen alone at randomization, but not among those receiving no respiratory support [37]. Administration of hyperimmune anti-COVID-19 intravenous immunoglobulin in severe and critical COVID-19 patients increased the chance of survival and reduced the risk of disease progression [38]. Early initiation of awake prone positioning in acute hypoxemic respiratory failure secondary to COVID-19 improved 28-day survival [39]. Discontinuation of renin–angiotensin system inhibition in COVID-19 had no significant effect on the maximum severity of COVID-19 but lead to a faster and better recovery [40].

Balbi [32] found that in emergency patients, parenchymal opacities most frequently did not show either a peripheral or central distribution (65%) or were peripherally located (30%), and bilateral lung involvement was found in 93% of cases. These are both in accordance with what we found; in our severe cases, the opacities had neither a peripheral nor a central distribution in 58.33% of cases in group 1, and, less frequently, in 33.33% cases in group 2. Bilateral lung involvement was found in most of our cases, 91.67% in group 1 and 100% in group 2.

Zheng [27] described a small amount of pleural effusion in patients with critical cases of COVID-19. We also found pleural effusion, in small to medium quantity, but in only 8.33 and 11.11% of patients in groups 1 and 2, respectively, which means few critical patients. The patient in group 2 might have had pleural effusion due more to major liver and kidney deterioration, and less to the extent of pulmonary involvement, though we cannot demonstrate this, since pleural fluid examination after biopsy proved inconclusive.

In critically ill patients in the ICU, although not COVID-19 patients, chest X-ray scores were found to be independently associated with length of stay [34]. This was not true for our COVID-19 patients; we found no correlation between any Brixia score and the number of days of hospitalization, nor was there any significant difference in the number of days of hospitalization between the two groups. Still, there was a significant difference in the number of days from admission until delivery: patients in group 1 delivered on the first day after hospital admission, whereas less than half the patients in group 2 delivered on the eighth day after hospital admission, and the rest continued their pregnancy normally, which shows the huge progress in COVID-19 health care and medication.

Chi [41] reported a total of 5.19% of all COVID-19-positive pregnant women who received mechanical ventilation, compared to 22 to 25% of critically ill pregnant patients in our study. Mechanical ventilation in pregnant women poses many challenges, including an increased risk of intubation failure due to increased airway edema, decreased functional residual capacity, and a higher risk of aspiration [42,43]. Trahan [42] reported that 33.33% of critically ill women who required mechanical ventilation were discharged home without requiring emergency delivery. We found a similar percentage of intubated patients; still,

in group 1, two of three patients required emergency cesarean delivery, one because of a dead fetus and the other because of worsening of maternal and fetal conditions, while the third delivered vaginally, so all intubated patients in group 1 required emergency delivery. The same was found in group 2, with one intubated pregnant patient who underwent an emergency cesarean section and one intubated non-pregnant patient. In group 1, the maternal condition improved after delivery in all cases; still, only one patient (33.33%) was safely released to home care, while between the other two, one required admission to the nephrology department because of kidney failure and one to the pulmonary disease department for continuing lung healing. In group 2, the patients' condition improved in both intubated cases, and they were safely discharged to home care. The results of group 1 are in accordance with those of Hirschberg [44], who reported that 40% of previously intubated patients were released to home care after delivering their babies, while the remaining 60% remained in a stable but serious condition after delivery, requiring further ICU/hospital health care. On the other hand, in group 2, all previously intubated patients were safely discharged to home care.

Though there were preterm deliveries in both groups due to maternal deterioration, as reported in other studies [5,45–49], with a high cesarean rate, as also reported by others [45], patients in group 1 delivered either vaginally or by cesarean section, while all patients in group 2 required preterm cesarean delivery. Still, as stated above, less than half of the patients infected with the Delta variant (group 2) delivered preterm when critically ill, and the rest continued their pregnancy normally; this was significantly different from group 1, in which almost all patients delivered preterm. This is partly in accordance with what was reported by Limaye [50]: women with severe/critical disease were more likely to have cesarean delivery (35.5%) or preterm delivery <37 weeks (25.8%). We report 77.78 and 100% of deliveries by cesarean section in the two groups, respectively, and preterm delivery at 32 and 34 weeks for most patients in group 1, but for less than half of the patients in group 2. We report values closer to those of DiToro (85% cesarean prevalence) [51], Matar (76%) [52], and Eman (89.4%) [11]. Metz [53] also reported a 59.6% risk of cesarean birth and a 41.8% risk of preterm birth in severely critical pregnant COVID-19 patients, the latter in accordance with what we found in group 2.

In the United States, the proportion of pregnant patients with severe to critical illness increased in April 2021; the total number of cases, however, remained low, and as the Delta variant predominated locally, both the case volume and the proportion of severe or critical illnesses increased significantly ($p = 0.001$ for trend), with over a quarter of pregnant patients who were diagnosed between 29 August 2021 and 4 September 2021 requiring admission for severe or critical illness [54]. This was not true for us, since, as written above, the Brixia score in the two groups showed that for patients with the Alpha variant, there was slightly higher lung involvement on admission in spring than in autumn, and they were released home or sent to another hospital with slightly less lung involvement than patients with the Delta variant; still, there was no significant difference between these scores. Murphy [55], studying the Alpha variant, demonstrated that following a COVID-19 infection in pregnancy, there was no increase in the incidence of preterm birth or neonatal intensive care unit admission compared with 5-year hospital data, and maternal symptom status did not influence neonatal outcomes. This is in accordance with what we demonstrated, i.e., that the condition of neonates at birth, reflected by the Apgar score, was approximately similar in the spring and autumn groups, and birth weight was slightly higher in the autumn group, but without significant differences between the two groups.

As an early risk assessment tool predicting a severe course of COVID-19, a low qSOFA (quick sequential organ failure assessment) score cannot be used to assume short-term stable or noncritical disease status in COVID-19 [56]. Additionally, the standard qSOFA is a poor screening tool in the prediction of severe maternal morbidity in pregnant patients with infections, and thus a pregnancy-specific screening system, qSOFA-P, was shown to improve the prediction of severe maternal morbidity in pregnant women with severe infections [57]. Combining severity tools such as computed tomography severity score (CT-

SS), national early warning score (NEWS), and qSOFA improved the accuracy of predicting mortality in patients with COVID-19 [58]. A further study to combine the qSOFA score and the Brixia score might show interesting results.

The major drawback of this study was its small sample size, with 21 enrolled patients. Calculating statistical significance with n = 9 and n = 3 exposed the data to type 2 errors and statistical bias. We also acknowledge the likelihood of skewed data with small numbers. Further studies with more patients might be needed to prove the conclusion of this study. Second, the outcomes of disease from these variants in the Romanian population may not be similar to the outcomes in other populations around the globe.

## 5. Conclusions

There was no significant difference in maternal and fetal outcomes between the two groups, except for the number of patients released to home care and the number of days from admission until delivery. There was no correlation between any Brixia scores (H, L, A, E) and maternal or fetal outcomes in both groups.

**Author Contributions:** Conceptualization, R.C. and D.S.; data curation, D.S.; formal analysis, L.V.B.; investigation, R.C., I.P. and M.A.; methodology, R.S.; project administration, R.S.; software, L.V.B.; supervision, R.S.; validation, R.S.; visualization, D.S.; writing—original draft, I.P., M.A. and L.V.B.; writing—review and editing, D.S. All authors have read and agreed to the published version of the manuscript.

**Funding:** This research received no external funding.

**Institutional Review Board Statement:** This study was conducted according to the guidelines of the Declaration of Helsinki and approved by the Research Ethics Committee of the Elena Doamna Obstetrics and Gynecology University Hospital in Iasi (number 4, 2 April 2020).

**Informed Consent Statement:** Informed consent was obtained from all subjects involved in the study.

**Data Availability Statement:** All data are available from the corresponding author upon reasonable request.

**Conflicts of Interest:** The authors declare no conflict of interest.

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
