# Peer review of "Brixia Score in Outcomes of Alpha versus Delta Variant of Infection in Pregnant Critical COVID-19 Patients"

_2673-8937, doi:10.3390/ijtm2010007_

Round 1
Reviewer 1 Report
This is a very interesting and novel study where the authors have investigated the prognostic usefulness of the Brixia scoring system to predict clinical severity in the context of the strain of virus causing COVID-19 in pregnant women.
However, I recommend that the authors consider the following questions and comments to improve their manuscript.
Questions and comments for the authors:
- How did they determine the type of viral strain? Was this determined with RT-PCR? I understand the likelihood was that the month of infection means certain strains were more prevalent but how can they be certain? Even 1 patient moving from one group to another would change their results significantly.
- Do these patients comprise all the pregnant women that were admitted in the institution? Can their dataset be expanded?
- If the authors wish to determine the usefulness of the Brixia score to determine clinical severity and outcomes, does the strain matter? With such small numbers combining them might be better to draw meaningful conclusions.
- Have the authors considered using an objective measure of clinical severity such as a modified QSOFA score and determine if there was an associative relationship with the Brixia score?
- Ultimately, calculating statistical significance with N=9 and N=3 will expose the data to type 2 errors and statistical bias.
- The study results conclude that the Brixia score is not useful but as mentioned above this can be assessed differently. However, the main outcomes are that patients who were admitted to hospital in spring 2021 versus autumn 2021 spent more time in hospital and had their babies were born soon after admission. Most recent studies suggest that severe illness is more common with the delta strain in pregnancy. This is despite greater viral prevalence.
- The authors have discussed at length lung abnormalities with COVID-19 in the discussion, but their null hypothesis was true. It would be a good idea to discuss their significant findings and how these fit with the literature. They should also consider published epidemiological data on the outcomes of pregnant women infected with different SARS-CoV-2 strains from around the world, which contradict their main findings. They should also acknowledge the likelihood of skewed data with small numbers.
- In addition, their main findings may have also been due to the use of new clinical interventions and a change in clinical management – e.g RECOVERY trial, greater use of non-invasive ventilation. These need to be assessed and discussed.
Author Response
Response to Reviewer 1
Open Review
(x) I would not like to sign my review report
( ) I would like to sign my review report
English language and style
( ) Extensive editing of English language and style required
( ) Moderate English changes required
(x) English language and style are fine/minor spell check required
( ) I don't feel qualified to judge about the English language and style
|
Yes |
Can be improved |
Must be improved |
Not applicable |
|
|
Does the introduction provide sufficient background and include all relevant references? |
( ) |
(x) |
( ) |
( ) |
|
Is the research design appropriate? |
( ) |
(x) |
( ) |
( ) |
|
Are the methods adequately described? |
(x) |
( ) |
( ) |
( ) |
|
Are the results clearly presented? |
( ) |
(x) |
( ) |
( ) |
|
Are the conclusions supported by the results? |
( ) |
( ) |
(x) |
( ) |
Comments and Suggestions for Authors
This is a very interesting and novel study where the authors have investigated the prognostic usefulness of the Brixia scoring system to predict clinical severity in the context of the strain of virus causing COVID-19 in pregnant women.
However, I recommend that the authors consider the following questions and comments to improve their manuscript.
Questions and comments for the authors:
- How did they determine the type of viral strain? Was this determined with RT-PCR? I understand the likelihood was that the month of infection means certain strains were more prevalent but how can they be certain? Even 1 patient moving from one group to another would change their results significantly.
-No, it was genomic sequencing of viral samples, performed by The National Institute of Public Health nationwide. We added in Materials and Methods:
Viral strains were determined by genomic sequences of viral samples, performed by The National Institute of Public Health nationwide and released as weekly bulletins [19].
-as Reference: 19.https://www.cnscbt.ro/index.php/analiza-cazuri-confirmate-covid19/2948-s-02-informare-cazuri-cu-variante-care-determina-ingrijorare-voc/file, page 3, accessed January 27th, 2022
- Do these patients comprise all the pregnant women that were admitted in the institution? Can their dataset be expanded?
-No, these dataset comprise only the women admitted with severe form of COVID-19. We highlighted in Materials and Methods:
Patients who had mild or moderate forms of the disease and patients who were in the COVID-19 ICU because they had been just operated upon and were also positive for COVID-19 (RT-PCR test) were excluded from the study.
We also added:
Patients who were in the ICU because they had just delivered by cesarean section, but were negative for COVID-19, were also excluded from this study.
- If the authors wish to determine the usefulness of the Brixia score to determine clinical severity and outcomes, does the strain matter? With such small numbers combining them might be better to draw meaningful conclusions.
We added in Results, highlighted in yellow:
Though no significant difference between the Brixia score between the two groups, there was a significant difference in outcomes between the two groups: all patients were discharged to home care in group 2 (delta), while only one-third were discharged to home care in group 1 (alpha), P = 0.02. There was a significant difference (P = 0.022) between the two groups regarding the number of days from hospital admission until delivery. In addition, most patients infected with alpha variant delivered when critically ill, while only less than half of those infected with delta variant delivered when critically ill, and the rest continued their pregnancy normally. Therefore, Brixia score could not be used to predict the outcomes.
- Have the authors considered using an objective measure of clinical severity such as a modified QSOFA score and determine if there was an associative relationship with the Brixia score?
-We added in Discussion:
As an early risk assessment tool predicting a severe course of COVID-19, a low qSOFA (quick sequential organ failure assessment) score cannot be used to assume short-term stable or noncritical disease status in COVID-19. [Sven Heldt]. Also, the standard qSOFA is a poor screening tool in the prediction of severe maternal morbidity in pregnant patients with infections, so that a pregnancy-specific screening system, qSOFA-P, improved prediction of severe maternal morbidity in pregnant women with severe infections. [Sarah Cagino]. Combining severity tools such as computed tomography severity score (CT-SS), national early warning score (NEWS) and qSOFA improved the accuracy of predicting mortality in patients with COVID-19. [Gokhan Akdur]. A further study to combine qSOFA score and Brixia score might show interesting results.
And in References:
- Heldt, S.; Neuböck, M.; Kainzbauer, N.; Shao, G.; Tschoellitsch, T.; Duenser, M.; Kaiser, B.; Winkler, M.; Paar, C.; Meier, J.; Lamprecht, B.; Salzer. H. qSOFA score poorly predicts critical progression in COVID-19 patients. Wien Med Wochenschr. 2021 Jun 29;1-9. doi: 10.1007/s10354-021-00856-4. Online ahead of print.
- Cagino, S.; Burke, A.; Letner, D.; Leizer, J.; Zelig, C. Quick sequential organ failure assessment: modification for identifying maternal morbidity and mortality in obstetrical patients. Am J Perinatol. 2022 Jan;39(1):1-7. doi: 10.1055/s-0041-1735624. Epub 2021 Sep 28.
- Akdur, G.; DaÅŸ, M.; Badakci, O.; Akman, C.; SiddikoÄŸlu, D.; Akdur, O.; Akçali, A.; ErbaÅŸ, M.; ReÅŸorlu, M.; Beyazit, Y. Prediction of mortality in COVID-19 through combining CT severity score with NEWS, qSOFA, or peripheral perfusion index. Am J Emerg Med. 2021 Dec;50:546-552. doi: 10.1016/j.ajem.2021.08.079. Epub 2021 Sep 2.
- Ultimately, calculating statistical significance with N=9 and N=3 will expose the data to type 2 errors and statistical bias.
-We added in Discussion:
Calculating statistical significance with N=9 and N=3 exposed the data to type 2 errors and statistical bias.
- The study results conclude that the Brixia score is not useful but as mentioned above this can be assessed differently. However, the main outcomes are that patients who were admitted to hospital in spring 2021 versus autumn 2021 spent more time in hospital and had their babies were born soon after admission. Most recent studies suggest that severe illness is more common with the delta strain in pregnancy. This is despite greater viral prevalence.
We added in Discussion:
In United States, the proportion of pregnant patients with severe to critical illness increased in April 2021; the total number of cases, however, remained low, and as the delta variant predominated locally, both the case volume and the proportion of severe or critical illnesses increased significantly (P=.001 for trend), with over a quarter of pregnant patients who were diagnosed between August 29, 2021 and September 4, 2021 requiring admission for severe or critical illness. [Emily Adhikari]. This was not true for us, since, as written above, the Brixia score in the two groups showed that for patients with alpha variant, there was slightly higher lung involvement on admission in spring than in autumn, and they were released home or sent to another hospital with slightly less lung involvement than patients with delta variant; still, there was no significant difference between these scores. Claire Murphy, studying the alpha variant, demonstrated that following a COVID-19 infection in pregnancy, there was no increase in the incidence of preterm birth or neonatal intensive care unit admission compared with 5-year hospital data, and maternal symptom status did not influence neonatal outcomes. This was in accordance to what we demonstrated, that the condition of neonates at birth, reflected by the Apgar score, was approximately similar in the spring and autumn groups, and birth weight was slightly higher in the autumn group, but without significant differences between the two groups.
- The authors have discussed at length lung abnormalities with COVID-19 in the discussion, but their null hypothesis was true. It would be a good idea to discuss their significant findings and how these fit with the literature. They should also consider published epidemiological data on the outcomes of pregnant women infected with different SARS-CoV-2 strains from around the world, which contradict their main findings.
-We did write, as mentioned above:
In United States, the proportion of pregnant patients with severe to critical illness increased in April 2021; the total number of cases, however, remained low, and as the delta variant predominated locally, both the case volume and the proportion of severe or critical illnesses increased significantly (P=.001 for trend), with over a quarter of pregnant patients who were diagnosed between August 29, 2021 and September 4, 2021 requiring admission for severe or critical illness. [Emily Adhikari]. This was not true for us, since, as written above, the Brixia score in the two groups showed that for patients with alpha variant, there was slightly higher lung involvement on admission in spring than in autumn, and they were released home or sent to another hospital with slightly less lung involvement than patients with delta variant; still, there was no significant difference between these scores. Claire Murphy, studying the alpha variant, demonstrated that following a COVID-19 infection in pregnancy, there was no increase in the incidence of preterm birth or neonatal intensive care unit admission compared with 5-year hospital data, and maternal symptom status did not influence neonatal outcomes. This was in accordance to what we demonstrated, that the condition of neonates at birth, reflected by the Apgar score, was approximately similar in the spring and autumn groups, and birth weight was slightly higher in the autumn group, but without significant differences between the two groups.
They should also acknowledge the likelihood of skewed data with small numbers.
-We added, in Discussion:
The major drawback of the study was its small sample size, with 21 enrolled patients. Calculating statistical significance with n=9 and n=3 exposed the data to type 2 errors and statistical bias. We also acknowledge the likelihood of skewed data with small numbers. Further studies with more patients might be needed to prove the conclusion of the study. Second, the outcomes of disease from these variants in the Romanian population may not be similar to the outcomes in other populations around the globe.
- In addition, their main findings may have also been due to the use of new clinical interventions and a change in clinical management – e.g RECOVERY trial, greater use of non-invasive ventilation. These need to be assessed and discussed.
-We added in Discussion:
There were protocol changes [Daniel Foreman] [34] and huge progress in clinical care of severe COVID-19 patients during the year 2021. In patients hospitalized with Covid-19, the use of dexamethasone resulted in lower 28-day mortality among those who were receiving either invasive mechanical ventilation or oxygen alone at randomization but not among those receiving no respiratory support [RECOVERY] [35]. Administration of hyperimmune anti-COVID-19 intravenous immunoglobulin in severe and critical COVID-19 patients increased the chance of survival and reduced the risk of disease progression [Saukat Ali] [36]. Early initiation of awake prone positioning in acute hypoxemic respiratory failure secondary to COVID-19 improved 28-day survival [Ramadeep Kaur] [37]. Discontinuation of renin-angiotensin system inhibition in COVID-19 had no significant effect on the maximum severity of COVID-19 but lead to a faster and better recovery [Axel Bauer] [38].
Submission Date
26 December 2021
Date of this review
26 Jan 2022 17:20:34

Reviewer 2 Report
I suggest that the paragraph starting with "The patients' age range was 25-44 years in group..." in the methodology section of the article should be changed to the first paragraph of the Results section.
The authors used statistical analysis, but it was not specified which program they used and which method they used. The "statistical analysis" section should be added to the last paragraph of the methodology section. Since the number of patients in this study was low, mean+SD should definitely not be used. It is appropriate to use the median (IQR) in this study.
Looking at the title of the article, it is understood that all the patients included in the study were pregnant. however, when Table-1 is examined, it is understood that non-pregnant patients were included in the study. Therefore, the methodology part of the article should be written in a clear language.
In addition, it would be appropriate to add a short paragraph about the development of the Covid-19 disease in the introduction part of the study. I recommend that authors benefit from the following articles on this subject:
Sahin TT, Akbulut S, Yilmaz S. COVID-19 pandemic: Its impact on liver disease and liver transplantation.
WorldJ Gastroenterol. 2020 Jun 14;26(22):2987-2999.
Altunisik Toptan S, Bayindir Y, Yilmaz S, Yalçınsoy M, Otlu B, Kose A, Sahin TT, Akbulut S, Isik B, BaÅŸkiran A, Koc C. Short-term experiences of a liver transplant center before and after the COVID-19 pandemic.
Int J Clin Pract. 2021 Oct;75(10):e14668.
I would like to reevaluate the article after necessary revisions are made.
Author Response
Response to Reviewer 2
Open Review
(x) I would not like to sign my review report
( ) I would like to sign my review report
English language and style
( ) Extensive editing of English language and style required
( ) Moderate English changes required
( ) English language and style are fine/minor spell check required
(x) I don't feel qualified to judge about the English language and style
|
Yes |
Can be improved |
Must be improved |
Not applicable |
|
|
Does the introduction provide sufficient background and include all relevant references? |
( ) |
(x) |
( ) |
( ) |
|
Is the research design appropriate? |
( ) |
(x) |
( ) |
( ) |
|
Are the methods adequately described? |
( ) |
(x) |
( ) |
( ) |
|
Are the results clearly presented? |
( ) |
(x) |
( ) |
( ) |
|
Are the conclusions supported by the results? |
(x) |
( ) |
( ) |
( ) |
Comments and Suggestions for Authors
I suggest that the paragraph starting with "The patients' age range was 25-44 years in group..." in the methodology section of the article should be changed to the first paragraph of the Results section.
-We moved that paragraph, as follows, highlighted in RED:
- Results
The patients’ age range was 25-44 years in group 1 (alpha, spring group) and 27-52 years in group 2 (delta, autumn group). The mean age and percentage of pregnant patients were not significantly different between the two groups (P = 0.156 and 0.553, respectively) (Table 1). Both groups consisted of mostly pregnant patients and a few non-pregnant patients (Table 1).
Table 1. Patient characteristics: mean values (± standard deviation) on upper line, and median values (quartile 1, quartile 3) on line below.
|
Patients |
Group 1 (Alpha variant) (n = 12) |
Group 2 (Delta variant) (n = 9) |
P-value |
|
Age |
32.75 (±6.26) 31.50 (27.5, 37.00) |
36.56 (±7.19) 37.00 (31.50, 39.50) |
0.175 |
|
Pregnant patients |
11 (91.66%) |
7 (77.77%) |
0.553 |
The Brixia score was evaluated on the chest X-ray as follows: each lung was divided into three regions, from apex to base, and changes in the lungs were noted for each of the six regions as follows: normal (0), interstitial changes (1), interstitial and alveolar changes with interstitial predominance (2), and interstitial and alveolar changes with alveolar predominance (3). The resulting total Brixia score could be from 0 to 18. [15]. We studied the highest score (H-score), corresponding to the only score for patients who had only one chest X-ray and the highest score for those with multiple X-rays, and the lowest score (L-score), the score assigned at admission (A-score) and the last score obtained before hospitalization ended (E-score) or the patient died [15]. All X-rays were evaluated and scored by two independent investigators (RC and RS) blinded to the clinical data. Any differences were reconciled. Interobserver agreement was 0.94.
The Brixia score in the two groups varied as follows (Table 2): for patients with alpha variant, there was slightly higher lung involvement on admission in spring than in autumn, and they were released home or sent to another hospital with slightly less lung involvement than patients with delta variant; still, there was no significant difference between these scores (Table 2).
Table 2. Brixia score in two groups of patients: mean values (± standard deviation) on upper line, and median values (quartile 1, quartile 3) on line below.
|
Patients’ Brixia score |
Group 1 (n=12) |
Group 2 (n=9) |
P-value |
|
Brixia H-score |
11.50 (±5.50) 13.00 (8.00, 16.00) |
11.67 (±3.16) 12.00 (9.50,14.00) |
.72 |
|
Brixia L-score |
7.14 (±3.23) 6.00 (5.00,10.00) |
7.00 (±5.35) 7.50 (1.75,11.75) |
.77 |
|
Brixia A-score |
11.57 (±4.39) 13.00 (7.00,15.00) |
10 (±6.97) 11.50 (2.75,15.75) |
1 |
|
Brixia E-score |
9.00 (±4.24) 8.00 (5.00,13.00) |
8.50 (±5.68) 11.00 (2.75,11.75) |
.84 |
H-score: highest score, corresponding to only score for patients with only one chest X-ray and highest score for those with multiple X-rays; L-score: lowest score; A-score: score assigned at admission; E-score: last score obtained before hospitalization ended or patient died.
The authors used statistical analysis, but it was not specified which program they used and which method they used. The "statistical analysis" section should be added to the last paragraph of the methodology section. Since the number of patients in this study was low, mean+SD should definitely not be used. It is appropriate to use the median (IQR) in this study.
-We highlighted in RED the statistics programme description, as the last paragraph of the methodology section:
Statistical analysis was performed with SPSS version 18 software (SPSS Inc., Chicago, IL, USA). For descriptive measures, we computed the mean, standard deviation, median, and quartiles 1 and 3 (because the variables follow a non-normal distribution). Therefore, to compare the data, the nonparametric Mann–Whitney U test was applied. The standard significance cut-off at P = 0.05 was used to determine hypothesis conclusion.
Looking at the title of the article, it is understood that all the patients included in the study were pregnant. however, when Table-1 is examined, it is understood that non-pregnant patients were included in the study. Therefore, the methodology part of the article should be written in a clear language.
-We added in Materials and Methods, highlighted in RED:
Patients admitted to the COVID-19 intensive care unit (ICU) of Elena Doamna Obstetrics and Gynecology University hospital in Iasi between January 1 and December 1, 2021, who had a critical form of the disease and underwent conventional chest X-rays, were included and retrospectively studied. Most of them were pregnant patients (n=18) and few were non- pregnant female patients (n=3). Patients who had mild or moderate forms of the disease and patients who were in the COVID-19 ICU because they had been just operated upon and were also positive for COVID-19 (RT-PCR test) were excluded from the study. Patients who were in the ICU because they had just delivered by cesarean section, but were negative for COVID-19, were also excluded from this study.
In addition, it would be appropriate to add a short paragraph about the development of the Covid-19 disease in the introduction part of the study. I recommend that authors benefit from the following articles on this subject:
Sahin TT, Akbulut S, Yilmaz S. COVID-19 pandemic: Its impact on liver disease and liver transplantation.
WorldJ Gastroenterol. 2020 Jun 14;26(22):2987-2999.
Altunisik Toptan S, Bayindir Y, Yilmaz S, Yalçınsoy M, Otlu B, Kose A, Sahin TT, Akbulut S, Isik B, BaÅŸkiran A, Koc C. Short-term experiences of a liver transplant center before and after the COVID-19 pandemic.
Int J Clin Pract. 2021 Oct;75(10):e14668.
-We added, highlighted in RED:
- Introduction
Severe pulmonary disease caused by the novel coronavirus [severe acute respiratory syndrome coronavirus 2 (SARS-CoV-2)], has devastated many countries around the world and has overwhelmed the medical system [1] The priorities of many institutions have changed to manage critically ill coronavirus infectious disease-2019 (COVID-19) patients [2].
And in References:
References
- Sahin, T.T.; Akbulut, S.; Yilmaz, S. COVID-19 pandemic: its impact on liver disease and liver transplantation. World J Gastroenterol. 2020 Jun 14;26(22):2987-2999. doi: 10.3748/wjg.v26.i22.2987.
- Altunisik Toplu, S.; Bayindir, Y.; Yilmaz, S.; Yalçinsoy, M.; Otlu, B.; Kose, A.; Sahin, T.T.; Akbulut, S.; Isik, B.; BaÅŸkiran, A.; Koc, C. Short-term experiences of a liver transplant centre before and after the COVID-19 pandemic. Int J Clin Pract. 2021 Oct;75(10):e14668. doi: 10.1111/ijcp.14668. Epub 2021 Aug 5.
I would like to reevaluate the article after necessary revisions are made.
Submission Date
26 December 2021
Date of this review
25 Jan 2022 23:30:59

Round 2
Reviewer 1 Report
The authors have kindly amended their manuscript in line with my comments.
Author Response
Response 2 to Reviewer 1
Open Review
(x) I would not like to sign my review report
( ) I would like to sign my review report
English language and style
( ) Extensive editing of English language and style required
( ) Moderate English changes required
(x) English language and style are fine/minor spell check required
( ) I don't feel qualified to judge about the English language and style
|
Yes |
Can be improved |
Must be improved |
Not applicable |
|
|
Does the introduction provide sufficient background and include all relevant references? |
(x) |
( ) |
( ) |
( ) |
|
Is the research design appropriate? |
( ) |
(x) |
( ) |
( ) |
|
Are the methods adequately described? |
( ) |
(x) |
( ) |
( ) |
|
Are the results clearly presented? |
( ) |
(x) |
( ) |
( ) |
|
Are the conclusions supported by the results? |
(x) |
( ) |
( ) |
( ) |
Comments and Suggestions for Authors
The authors have kindly amended their manuscript in line with my comments.
Submission Date
26 December 2021
Date of this review
04 Feb 2022 13:56:06
Thank you very much!
Roxana Covali

Reviewer 2 Report
Thank you
Author Response
Response 2 to Reviewer 2
Open Review
(x) I would not like to sign my review report
( ) I would like to sign my review report
English language and style
( ) Extensive editing of English language and style required
( ) Moderate English changes required
(x) English language and style are fine/minor spell check required
( ) I don't feel qualified to judge about the English language and style
|
Yes |
Can be improved |
Must be improved |
Not applicable |
|
|
Does the introduction provide sufficient background and include all relevant references? |
(x) |
( ) |
( ) |
( ) |
|
Is the research design appropriate? |
(x) |
( ) |
( ) |
( ) |
|
Are the methods adequately described? |
(x) |
( ) |
( ) |
( ) |
|
Are the results clearly presented? |
(x) |
( ) |
( ) |
( ) |
|
Are the conclusions supported by the results? |
(x) |
( ) |
( ) |
( ) |
Comments and Suggestions for Authors
Thank you
Submission Date
26 December 2021
Date of this review
02 Feb 2022 07:04:00
Thank you very much!
Roxana Covali
